# An Overview of Hospital Capacity Planning and Optimisation

**DOI:** 10.3390/healthcare10050826

**Published:** 2022-04-29

**Authors:** Peter Humphreys, Belinda Spratt, Mersedeh Tariverdi, Robert L. Burdett, David Cook, Prasad K. D. V. Yarlagadda, Paul Corry

**Affiliations:** 1School of Mathematical Sciences, Queensland University of Technology, Brisbane, QLD 4000, Australia; sprattbelinda@gmail.com (B.S.); r.burdett@qut.edu.au (R.L.B.); y.prasad@qut.edu.au (P.K.D.V.Y.); p.corry@qut.edu.au (P.C.); 2World Bank, Washington, DC 440236, USA; mtariverdi@worldbank.org; 3Princess Alexandra Hospital, Brisbane, QLD 4000, Australia; david.cook@health.qld.gov.au

**Keywords:** hospital capacity and planning, optimisation, literature review, overview, health care, holistic, hospital

## Abstract

Health care is uncertain, dynamic, and fast growing. With digital technologies set to revolutionise the industry, hospital capacity optimisation and planning have never been more relevant. The purposes of this article are threefold. The first is to identify the current state of the art, to summarise/analyse the key achievements, and to identify gaps in the body of research. The second is to synthesise and evaluate that literature to create a holistic framework for understanding hospital capacity planning and optimisation, in terms of physical elements, process, and governance. Third, avenues for future research are sought to inform researchers and practitioners where they should best concentrate their efforts. In conclusion, we find that prior research has typically focussed on individual parts, but the hospital is one body that is made up of many interdependent parts. It is also evident that past attempts considering entire hospitals fail to incorporate all the detail that is necessary to provide solutions that can be implemented in the real world, across strategic, tactical and operational planning horizons. A holistic approach is needed that includes ancillary services, equipment medicines, utilities, instrument trays, supply chain and inventory considerations.

## 1. Introduction

### 1.1. Context

Hospital capacity is defined in a general sense as an upper bound that describes the best possible performance of the hospital in terms of productivity, output or number of patients treated [1]. This paper seeks to provide an overview of the optimisation of hospital capacity and planning, and its focus will be to take a detailed view, mapping out its various components.

All hospitals are constrained by their available resources and the public, for the most part, have limited funds to avail themselves of those services. For example, approximately half the world’s population do not have access to basic health care [2]. The demand for hospital services exceeds capacity at a global level [3].

One of the major challenges in any form of research is the practical application of findings in the real world. There is a need to link academic research and optimisation models to the day-to-day operational needs of hospitals [4]. When managing hospitals, planners and executives must contend with many challenging capacity-related questions.

The following is a small snapshot:What proportion of time should be allocated to different specialties in operating theatres?How many ward beds should be allocated for each specialty?What is the impact of changes to the master surgical schedule on capacity utilisation throughout the hospital?How well aligned is the current hospital configuration to forecasted patient case mix and volume?What improvements would result from a proposed expansion or reconfiguration?What are the benefits of outsourcing or caseload sharing applied at a regional level?

This multi-faceted state of flux gives rise to the necessity for a hospital system to constantly adapt to optimise its objectives and performance goals and to deliver the best health care for everyone.

### 1.2. Significance and Scope

Many of the world’s health systems are struggling to maintain financial sustainability in an uncertain and changing environment. Examples of contributing factors include expanding and aging populations; increasing numbers of people with chronic, long-term conditions; costly infrastructure and medical technology investments (compounded by low levels of capital spending over many years); rising labour costs and staff shortages; and growing demand for a larger ecosystem of services (general practitioner, community and home based, mental health, long term, etc.) [5]. Global health care spending could reach over $10 trillion by 2022 and is one of the fastest-growing industries in the world [2]. PolicyAdvice [2] goes on to say that the internet of things (training, patient monitoring, preventative care, and workflow optimisation) and digitalisation is set to revolutionise health care permanently.

So why is there a need for yet another review paper on hospital capacity? As we will later show, there are many opportunities to improve the health care sector, and to apply advanced operations research methods. It is quite evident that the health care sector is not optimised sufficiently and may well never be. In a report by Limb [6], “Health economists have estimated that a fifth of spending on healthcare in countries of the Organisation for Economic Co-operation and Development is ineffective and have issued a call for action”. Even if there were methods to fully optimise hospitals today, due to the rapid, ever-changing environment especially with respect to digital technologies, government policy and medical procedure advancement, there is a very strong need for an overarching framework for hospital capacity planning and optimisation which brings together all the parts of the hospital. Secondly, there is a need for optimisation and mathematical decision support tools to be developed that can operate within this framework, that are novel, implementable, and extendable.

## 2. Overview

### 2.1. Methodology of the Overview and Mapping Process

This paper was designed to be a combination of the overview and mapping review types as defined by [7]. Overviews attempt to survey the literature, while a mapping review attempts to map out and categorize existing literature, identifying gaps leading to further reviews and/or research [7]. The first part of this article’s review process was to analyse the entire hospital system and understand it. In addition to analysing the literature, interviews with staff from South-East Queensland hospitals were conducted (South-East Queensland is a large Australian metropolitan region). The interviews and site tours were conducted with managers from the following departments: emergency, outpatient clinics, surgical bookings, intensive care unit, day surgery admission, imaging, operating theatres, equipment, storerooms, central sterile services department, allied health, pre-op and recovery wards. Figure 1 shows the concept map that was constructed to illustrate how the parts of the hospital system interact and relate to each other. This map was employed as a framework to search for the literature review. A table was developed to document the search process for all the parts of the hospital system (see Table 1). Due to the complex nature of the subject, many articles were relevant to multiple parts of the system. A search of the top medical journals was also conducted to ensure the literature review was targeted to the industry. The QUT and Google Scholar search engines were used. The QUT search engine has access to over 400 databases including over 100 specifically related to health. Another table was developed to document every article included in this paper to facilitate analysis over several key parameters including mathematical methodologies employed, date of research, authors, parts, and sub-parts of the hospital. The purpose of choosing this style of research approach was to achieve transparency and robustness, with an aim to establish past and future trends, and to identify the gaps in the literature.

### 2.2. Overview of Hospitals

A brief overview of the hospital system is given in this section. A hospital is an institution providing medical and surgical treatment and nursing care for sick or injured people. The first hospital in the modern sense of the word was developed in the 4th century, by a wealthy Christian widow named St. Fabiola in Rome [8]. Since then, the hospital system has continually evolved. A consideration of hospital optimisation starts with government policy, human philosophy of care and stewardship of resources.

A hospital is a complex system made up of many parts. The following gives a snapshot of some of the parts of a hospital and how they relate to each other within the context of the whole system. At the highest level, hospitals are places where activities requiring resources (e.g., operations and medications) are performed on patients. There are many different patient care pathways and specialties. Within each of these pathways and specialties, many intricate and valuable pieces of equipment are required. There are financial limitations on resources; staff are qualified for specified tasks and have constraints as to when and where they can work [9]. Methodologies such as staff and patient scheduling, administration, queuing, information and filing systems contribute immensely to hospital efficiency. Aside from the core activities, there are context-specific variations between countries. Patients have diverse backgrounds, with various religious, ethnic, and socio-economic needs and desires. Ancillary parts of the hospital provide essential support services to the core activities of the hospital. Buildings and equipment require regular maintenance and must be managed well to avoid excessive costs and operational bottlenecks [10]. Consumables such as face masks, hand sanitiser, food, instruments, and linen are required at the right time in the right quantities. Supply chains that deliver these consumables need to be risk diversified as the recent COVID-19 pandemic has shown. Visitors need car parks, public transport, coffee shops and chaplains. Even the architectural design of the hospital and location of various wards has a significant impact on cost and efficiency [11]. Energy, heating, and water consumption are precious resources that need to be carefully used and optimised [12]. Waste processing and recycling can also have a significant effect on the hospital system that can affect the financial budget and therefore total number of services performed [13]. Overarching all these parts of the hospital is legislation and philosophy of care. This realm of management has an enormous impact on every part of the hospital and is ultimately responsible for how the hospital is run.

In summary, just as the human body is made up of many parts, the hospital system is synonymous with one body made up of many parts. Each part is important to the overall function and is intricately related to each other. Therefore, a study into optimising an entire hospital needs to consider all parts of the system and how they relate to each other. Hospital capacity optimisation is much more than solving a mathematical programming problem because it involves the subjective factors in addition to the objective ones. For example, themes such as teamwork, trust, social interdependence, and communication have a major impact on productivity [14]. Story [9] goes further and asserts that culture, especially in health care, is perhaps the dominant detractor to true capacity optimisation. Figure 1 graphically displays the parts of a hospital system and how they relate to each other. For reference, ‘External uncontrollable’ includes aspects such as location, supply chains, wars, natural disasters, and pandemics. ‘External controllable’ refers to location, supply chains, car parking, transportation, waste, recycling, playgrounds, and green spaces.

The diagram has been intentionally laid out to represent a physical hospital building. Just as a building has foundations, a hospital organisation is founded on philosophy, policy, and management principles, e.g., What is the purpose of the hospital? Why do we look after the sick? Emanating from philosophy comes the architecture and design of the hospital, both physical and operational. These two parts of the hospital are coloured dark grey to denote that they are foundational and cornerstones of the hospital. All other parts of the hospital are affected by these parts. The light grey parts represent the remainder of the hospital, and the arrows describe the relationships between each of the parts. The position denotes whether they exist inside or outside the hospital building. The roof of the building is shaded differently as it represents the collection of optimising elements that apply to each part of the hospital. For example, consider optimising the operating theatre resources of the hospital. There is a controllability element to optimising that—is it easy to change its capabilities? There is a timeliness element—how long will it take to change its capabilities? There is a probability or distribution element to its optimisation—what is the probability of surgery cancellation or finishing late? There is a rate of change element—when will I need to change the operating theatre capabilities when patient case mix changes and population size grows in the future? Finally, there is a financial element to the optimisation of the operating theatre. All these optimisation elements need to be considered within each part of the hospital and how they relate together within the context of entire system. They also need to be understood from the perspective of each planning horizon—strategic (long term), tactical (<1 year) and operational (day of operations). The framework (see Figure 1) is proposed here as an aid to facilitate an understanding of these matters as they relate to hospital capacity optimisation and planning.

### 2.3. Previous Reviews

This section summarises the review papers that have already been published regarding hospital capacity optimisation. For succinctness, Table A1 summarises the papers by year and part of the hospital and their key findings. This appendix can be used as a reference point to compare with articles found in this article’s review.

Fifteen review papers were found over the last twenty-one years. Seven were classified as core activities, three were classified as philosophy, two as external uncontrollable, and one paper was classed as patient case mix. Only two papers were classed as holistic, meaning that they looked at the overall hospital system. One focussed on capacity strain (i.e., approaching capacity limits) issues and inpatient outcomes [15] while the other had a specific focus with a review of ten papers on implementing process-oriented organisation designs [16]. This literature review is different from these papers because it summarises all the parts of the hospital from an operations research perspective, with a unique focus on problem-solving methodologies related to hospital capacity and planning.

### 2.4. Literature Analysis of the Parts of a Hospital

The literature search identified over 3000 articles and 245 of these were included for further comment and analysis in this article. A systematic approach was applied to decide which articles to include or not include. Based on a qualitative assessment by the authors of this review, any articles included relate to hospital capacity planning and optimisation within an operations research context and judged to come from reputable sources. These articles are grouped into the parts of the hospital identified in Figure 1. It is worth noting that it was not the intention of this research to include every article written on the subject, but rather to include a representative cross-section of literature, so that a high-level analysis of the subject may be discerned, for practical reasons. If less than approximately 10 papers were found for a part of the hospital and methodology type, then all the papers were included, but if more than approximately 10 papers were found, then a representative sample was included based on a subjective assessment of the contribution of the article. Additionally, note that not all 245 articles were cited in this paper but were used for the high-level analysis. This section firstly presents the articles in various graphs to draw out relationships and trends, then it goes on to analyse each part of the hospital in detail for key papers.

Table 2 summarises the papers found by part, and sub-part of the hospital, and the approach taken. Many articles did not have a mathematical approach, but rather used a qualitative research approach, e.g., process oriented, experimental, or survey based. This table highlights the fact that not all parts of the hospital have been researched extensively. For example, only five articles were found regarding architecture and hospital capacity optimisation and planning. It also highlights that stochastic approaches are employed more abundantly than deterministic ones. For the purposes of this classification, stochastic refers to approaches that consider random variables as opposed to static deterministic ones.

Figure 2 displays a summary of the articles regarding the mathematical methodology employed by part of the hospital. It clearly demonstrates that the bulk of research has focussed on the core operational workings of the hospital, with a noticeable lack of attention to the ancillary parts of the hospital, namely ancillary, architecture, external controllable, human resources, operational systems, patient case mix, philosophy, pre- and post-hospital considerations and holistic. It also highlights a noticeable disparity between methodologies employed. A three-dimensional chart was chosen to illustrate the magnitude of articles for each category.

Figure 3 shows that stochastic approaches are more common than deterministic approaches for every part of the hospital.

There are pros and cons to every problem-solving method. When solving a complex problem, a common strategy employed by researchers is to first solve a simplified version or variant, and then incrementally add complexity. Throughout this iterative journey, various methods may be employed, and the benefit of this strategy is that a thorough, robust understanding of the system is obtained. For example, a deterministic integer programming approach may be used initially, and then a stochastic simulation model might be used to understand the uncertain components of the system. This methodology was employed by Banditori et al. [17]. This observation may account for the reason why we see such a spread of methodologies across the various parts of the hospital.

Now that the contextual overview of the articles has been made, each part of the hospital system will be analysed in detail to draw out further insights.

#### 2.4.1. Architecture and Design

There has been very little research on architecture and design with respect to hospital capacity optimisation. Optimising facility-related processes in hospitals has the potential to achieve major savings and improve medical processes at the same time—meeting the strategic need to reduce health care costs without having a negative impact on the quality of the core competencies and processes of hospitals [18]. That paper presents the findings of the Optimisation and Analysis of Processes in Hospitals research project, which analysed the interaction between primary (medical) and secondary (facility management) business processes in six hospitals, with a view to identifying a holistic approach and comprehensive framework for evaluating business processes to ensure their optimisation. It follows that any study to determine an optimal hospital capacity must include optimising facility-related processes as an integrated hospital activity.

Diez and Lennerts [19] investigated the interdependencies between facility management performance, costs, and primary processes in hospitals. They developed an activity-based costing model to assign facility management costs to a unit within a hospital and acknowledged that the model needed to be extended to the rest of the hospital. Pati et al. [20] states that while facility design is increasingly playing a role addressing strategic organisational objectives, issues pertaining to facility maintenance have typically been left out of the decision-making process. The paper discusses two sets of facility maintenance indicators that have the potential to bridge the traditional divide between design and maintenance. This study demonstrates the need to incorporate maintenance issues into the overall strategic design process of a hospital.

However, articles found in the public domain have cited that architecture and design of hospitals are areas that need more attention. Heller [11] states that, in the past, architecture has typically not been considered a key predictor of health outcomes, but now it is changing as there is a growing awareness of the importance of incorporating design that promotes operational efficiency into the health care built environment. The article went on to say that there were many avoidable cases with surgical complications that lead to $5 billion in wasted spending each year. This article demonstrates the importance of design, not only to the normal operating paradigm, but to the surge operating paradigm for events such as pandemics and earthquakes. While there has been some research into the field of architecture design and its implications on the efficiency of hospitals, more research is needed in this area. The impact of good architecture design should be quantified by a model that has sufficient detail to account for the operational efficiencies that the design of a building affords. A conclusion from this review of literature is that there is a need for hospital developers to learn from and apply the research that is currently available.

#### 2.4.2. Ancillary Components

Highly related to the subject of architecture is the design of ancillary components of a hospital such as maintenance, green areas, cafés, playgrounds, energy systems, waste recovery and recycling, and clean water systems. No articles were found that discussed the implications of cleaning activities in hospital capacity optimisation, and yet without appropriate cleaning staff and resources, hospital efficiency would be impacted. Busby [21] did, however, model cleaning staff for bed change-over in her study on surge capacity, but did not account for variation, or other cleaning processes such as instrument tray sterilisation. In the New York Times, Cary [22] gave a case study of a Cholera Treatment Center in Haiti that demonstrated the impact that good design has on health care outcomes. The wastewater-treatment system was designed to prevent recontamination of the water table, stopping the spread of disease. Budak and Ustundag [13] developed a mixed-integer linear programming model to determine the optimal number and locations of the facilities for efficient waste management in health care by minimising the total cost. Through sensitivity analysis, this study established the necessity of various strategies for various waste amounts. It also demonstrates that waste management should be included in a study of entire hospital capacity optimisation. If waste management is not adequately accounted for when modelling entire hospital capacity, then staff utilisations may be underestimated, rendering throughput results inaccurate. Recently, there has been technological advancement in the field of energy, cooling, and heating [12,23,24,25]. Implementation of these solutions has the potential to save a significant amount of money which can be used in other areas of the hospital, while at the same time reducing supply risks of these critical resources.

#### 2.4.3. External Controllable

External controllable aspects of hospital include carparks, supply chains, waste management and recycling, playgrounds, green spaces, and location. Six papers were found that considered location elements of a hospital [26,27,28,29,30,31], one that focussed on supply chain issues [32], and two concerning multi-hospital networks [33,34]. Multi-hospital networks is a subject that has been alluded to within the context of case mix planning discussed earlier. It must also be noted that this subject has implications on the inventory models, supply chain models, surge demand events and resource sharing within hospitals. Mahar, Bretthauer and Salzarulo [33] developed an optimisation model to determine the best hospitals within a network to deliver a specialised service. The model considered financial criteria and patient service levels. Their results demonstrated the benefits of sharing resources and highlighted the need for optimisation in this domain. Similarly, Mestre, Oliveira and Barbosa-Póvoa [34] built two models pertaining to location and allocation to improve geographical access while minimising costs. Clearly, any optimisation on this topic will require context-specific information for the group of hospitals considered. There is a need for a framework to be developed that can be used as a generic guide for any network of hospitals to use, which incorporates all the components of a hospital that it relates to. It is hypothesised that several different optimisation techniques may be needed in the framework so that the best methodology can be used for any given context.

#### 2.4.4. External Uncontrollable

External uncontrollable factors include surge events, such as earthquakes, wars, and pandemics, supply chains, public transport and locations. Many articles found for this part of the hospital dealt with coping mechanisms and reallocation of hospital capacity amid surge events, primarily pandemics [35,36,37,38,39,40,41,42,43,44,45,46,47,48,49,50,51,52,53,54,55,56,57,58,59]. Surge events highlight the importance of modelling the detail of every element within the hospital, from the hand sanitiser required, to the availability of doctors and specialised equipment, and how they interact with each other. The recent COVID-19 pandemic has brought these issues to the forefront. While this subject has received comprehensive attention, more research is needed in the areas of supply chains and location issues.

Supply chain issues and management policies are generally not considered in hospital optimisation studies, but they are a critical part of the system. Bhakoo et al. [60] set out to study the inventory management practices of participants in health care supply chains. They found several contingent factors that influence the type of collaboration arrangement. These include product characteristics, spatial complexity, degree of goal congruence between supply chain partners, the role of the regulatory environment and the physical characteristics of the organisation. More research is needed in this area to determine how to optimise such a system. Niakan and Rahimi [61] developed a multi-objective mathematical model to optimise medicinal drug distribution to health care facilities. The model minimised total inventory and transportation costs, while minimising forecast error, thus mitigating product shortage and expired drug risks.

#### 2.4.5. Human Resources

Four articles were found concerning human resources. Lennerts, Abel, Pfründer and Sharma [18] surveyed registered nurses, physicians, and hospital executives about the shortage of nursing staff. Not surprisingly, substantial proportions of respondents perceived negative impacts on care processes, hospital capacity, and nursing practice, but there were also many areas of divergent opinion within and among these groups. The paper went on to say that resolving these differences was key to improving the quality and safety of patient care. Bornemann-Shepherd et al. [62] conducted a study aimed at improving safety and patient and staff satisfaction within the emergency department. This quality-improvement project involved a questionnaire to gain feedback from key staff and patients and a task force was setup to implement the measures. Articles of this type show, via feedback from those questionnaires that a multi-disciplinary approach is needed to comprehensively address the hospital capacity optimisation problem. Kaw et al. [63] demonstrated a nursing staff cost reduction of 7% by analysing past nurse absences and developing a mixed-integer program to minimise costs subject to the size and skill mix of a nursing resource team, overtime costs and casual staff costs.

The nurse scheduling problem (NSP) has been considered extensively. Approximately 2000 articles were returned from the literature search, but only three articles were included here to give a representative snapshot of the literature. Cheang et al. [64] gives a bibliography and summary of the research on the NSP, discussing methods. They conclude that while optimisation approaches may be found, the computational time can restrict the size and complexity of the problem solved. They also acknowledged that it was difficult to incorporate all the hard and soft constraints into these approaches, which has lead researchers to solve the NSP using meta-heuristics. The authors of [65] analysed the literature from a practical implementation point of view and noted that there was a disparity between the models and the practical requirements of the real world. Smet et al. [66] also agreed with this assertion and went on to propose a more realistic NRP, presenting a suite of hyper-heuristics able to provide solutions.

#### 2.4.6. Operational Systems

Operational systems are tools that are used to administrate the hospital. These include but are not limited to information technology (IT) systems, accounting packages, patient booking systems, audio-visuals and communication systems. Most of the literature for this part of the hospital related to IT systems. IT systems can have an enormous impact on hospital capacity. For example, electronic dashboards that increase communication across hospital departments [67]; efficient management of medical records [68]; web-based real-time capacity management [69,70]; and data capture to track health care more efficiently [69]. It is this level of detail which needs to be incorporated into hospital capacity estimations since it directly affects all parts of the hospital system. For example, if a model is being used to determine the most efficient way to increase capacity, then for a given process improvement measure, it needs to calculate the flow-on effect to downstream and upstream processes within the whole system. Some performance improvements for one component may have little or no benefit to overall system performance. Similarly, a performance improvement in one component may have a profound benefit to overall system performance. For example, Burdett and Kozan [71] found that patient activity end times, such as post-op, were not acted upon because the IT system did not alert staff. This lead to extended length of stay durations for patients, which in turn reduced overall system capacity. Similarly, Ahalt et al. [72] used discrete event simulation to determine the most appropriate emergency department crowding score for a given hospital. This alert system allowed managers to reallocate resources in response to impending demand, increasing efficiency and patient care.

#### 2.4.7. Core Activities

Perhaps one of the most important activities with respect to hospital capacity management is operating theatre scheduling. Building on from previous research [23,73,74,75,76], Spratt and Kozan [77] completed significant work in this area by demonstrating an integrated rolling horizon approach to increase efficiency. They used a constructive heuristic and two hyper-metaheuristics to produce near optimal feasible solutions. While they factor in many variables that affect the schedule, such as surgeon availability, their scope does not include ancillary staff such as nurses and cleaners, shared equipment, and upstream and downstream hospital processes that can cause major bottlenecks when modelled as an entire hospital system. Burdett and Kozan [71] addressed the scheduling problem using a sophisticated flexible job shop scheduling model. This was a valuable contribution because it addressed the upstream and downstream processes that many other articles did not consider. This study, however, did not model the emergency department and other shared resources used by other parts of the hospital. The authors of [78,79] incorporated equipment and all the staff types required in surgeries, but did not account for stochasticity, the patient treat-in-time objective and only considered a short planning horizon of a week or less. The other area for improvement could be to consider the difference between the estimated surgery time and the actual surgery time. If the plan is more accurate, then this would lead to reduced variability, reaction and noise within the system. Burdett and Kozan [80] suggested the strategic placement of idle time and the dynamic selection of multiple free resources at each time in the scheduling process, proved to be effective in addressing uncertainty and variability.

Nguyen et al. [81] defines the number of beds to be optimal when the unoccupied beds is not excessive, patients transferred due to a lack of beds available is not excessive, and there are several beds available for unscheduled admissions. Kokangul [82] used a combination of deterministic and stochastic approaches to optimise bed capacity in a hospital unit. Cochran and Roche [83] developed a capacity planning tool in Microsoft Excel, based upon queuing theory and financial data. Li et al. [84] developed a multi-objective decision support model for allocation of beds in a hospital. The model was based on queuing theory and goal programming to allocate beds to meet targets and objectives related to customer service and profits. Ben Abdelaziz and Masmoudi [85] developed a multi-objective stochastic model to minimise three functions: bed creation and management, doctors and nurses. Utley and Peters [86] estimated the number of beds required for the smooth operation of a multi-specialty paediatric intensive care unit. Their study considered variability in timing, nature of referrals and length of stay. Devapriya et al. [87] developed a decision support tool for planning and budgeting of current and future bed capacity, and evaluating potential process improvement efforts. The key inputs of this discrete event simulation model included timing, quantity and category of patient arrivals and discharges; unit-level length of care; patient paths; and projected patient volume and length of stay. While all these papers addressed the aspect of bed planning and allocation they were intended for, there were not any papers that optimised bed capacity within the context of the entire hospital system and all its constraints, for example shared equipment and staff resources, medical inpatients and outpatients, surgical inpatients and outpatients, emergency department, laboratories, imaging and allied health professionals.

McManus et al. [88] established that queueing theory accurately modelled the level of need for intensive care. They concluded that the stochastic nature of patient flow may falsely lead health planners to underestimate resource needs in busy intensive care units. Madell et al. [89] conducted a rapid review of literature focussing on optimisation of resource use in hospitals. They outlined several practical strategies that hospitals could employ to optimise their overall efficiency. Chief among these was the use of specially trained nurses for special patient cases. Khaiter et al. [90] tried to optimise the resource allocations for the image guided therapy department of a hospital. They found that none of the investigated optimisation algorithms was able to optimise the schedules with respect to all the selected time-based performance criteria. Each algorithm generated schedules which are more efficient from the perspective of a single performance indicator. It was suggested that for a complex non-trivial problem of this kind, a hybrid approach combining several optimisation techniques might be successful. Bastian et al. [91] optimised resources across the United States Military Hospital System using mixed-integer linear programming, including some stochastic elements, but this was only for staff and funding. Another study by Feng et al. [92] produced a multi-objective stochastic mathematical model for medical resource allocation in emergency departments. It allocated the resources (i.e., staff, equipment, and beds) to minimise length of stay and medical waste cost. This highlights the need to model hospital resources with a high level of detail. 

Ahmadi et al. [93] presented an up-to-date review of research in the field of inventory management of surgical supplies and instruments. They organised the papers into two groups: those published by scientific researchers who developed optimisation techniques and those that were published by practitioners and reported their observations of the current issues in the operating room. An interesting finding was that preference card optimisation (i.e., the items and their quantities) was a topic that had been untouched thus far in the research literature. Another question that remains unanswered in the literature is the location and the quantity of surgical supplies that must be stocked according to the operating room’s specific process. A further question that needs to be answered is what methodology can help physicians to decide the appropriate quantity of material to be opened before the procedure with the aim of minimising waste without sacrificing patients’ quality of care. Optimal surgical tray configuration is another topic that needs addressing. They state that the future research direction is to develop stochastic models which must consider both cost, service level, operational risk, and disruption risk. These findings were also echoed by the staff in the case study hospitals that confirmed that management of preference cards and surgical trays is a key issue—specifically around the timing and resources required to sterilise them so they are available for surgeries.

Emergency department optimisation has received significant attention in the literature. A literature review by Ahsan et al. [94] found that not all modelling approaches were suitable for all situations and there was no critical review of optimisation models used in hospital emergency departments. Their analysis of all the methodologies revealed that every modelling approach and optimisation technique has some advantages and disadvantages, and their application is also guided by the objectives.

As a general comment on the articles found, there was little or no consideration of integration with the rest of the hospital and the support staff such as cleaning, ward and administration. One article that did highlight the importance of ancillary tasks such as medication delivery and lab sample collection is Batt and Terwiesch [95]. They noted that a load dependent mechanism, where staff in an upstream stage proactively initiate tasks normally handled by downstream staff, lead to a 20 min reduction in treatment time. The emergency department needs to be optimised within the context of the entire hospital system since it is integrally related, using shared resources, and in many cases dependent on hospital beds being available for patients to leave the emergency department. A similar conclusion was drawn by Zhu, Fan, Yang, Pei and Pardalos [4], where they recommend a better integration of compatible resources and the need researchers to narrow the gap between theory and practice. Even ancillary infrastructure such as carparks can affect the number of people arriving at an emergency department if patients have a choice of hospitals. Lack of carparking can also delay patients arriving on time for appointments and be a leading contributor to staff turnover.

#### 2.4.8. Patient Case Mix

Freeman et al. [96] state that case mix planning refers to allocation of time to operating rooms and is an important tool for achieving the goals of a hospital with respect to quality of care and financial position. Hof et al. [97] performed the first standalone literature review of case mix planning. They state that, in general, literature on case mix planning is scarce. Three gaps in the research were identified. Firstly, the incorporation of stochastic parameters into strategic case mix planning problems. Secondly, the consideration of hospital systems as they increasingly face market and financial pressures. Thirdly, whether it is more cost efficient to provide the majority of services in one hospital or have many specialised hospitals that each focus on specific services.

Freeman, Zhao and Melouk [96] developed a multi-phase approach that used mathematical programming and simulation to generate a pool of candidate solutions to the case mix planning problem. Each candidate solution was evaluated with respect to a broad range of strategic and operational performance measures. In comparison to a more traditional single-solution approach, they found that the solution pool approach identified case mix plans with higher expected patient reimbursement, lower over-utilisation of operating theatre time, and lower variability in the number of beds required in downstream recovery wards. McRae et al. [98] analysed the effect of economies of scale and scope on the optimal case mix of a hospital or hospital system. The non-linear mixed-integer program they formulated, however, did not account for staffing resources and for variation within the system. McRae and Brunner [99] subsequently tried to address these limitations and developed a framework for evaluating the impact of uncertainty and the use of different aggregation levels in case mix planning.

#### 2.4.9. Philosophy, Policy and Management

Philosophy of care, policy, and management has an enormous impact on hospital capacity. This is illustrated in Figure 1 by positioning philosophy as the foundation upon which all other parts of the hospital depend. Durán and Wright [100] devote a chapter in their book to hospital governance and give an excellent historical synopsis of the various issues around the world and how they impact health care. Positions on abortions and euthanasia greatly affect the efficiency and throughput of a hospital. Respect for patient’s religious beliefs and preferences such as diet also have an impact. Zaric [101], in his book ‘Operations Research and Health Care Policy’, brings together a group of papers by leading experts, showcasing the current state of the field of operations research applied to health care policy.

There were five articles chosen for consideration relating to financial matters within hospitals. The authors of [102] state that time-driven activity-based costing (TDABC) has been suggested as the cost-component of value-based health care capable of addressing costing challenges. Only one paper proposed a multi-objective stochastic programming (MSP) model to maximise both revenue and equity for patients [103]. The rest ([102,104,105,106]) discuss static analysis of costs, though essential and inherently of critical importance, has limited use informing future dynamic operating paradigms proposed by strategic and tactical models. Clearly, there exists the need for further research to incorporate financial analysis into a multi-objective holistic optimisation of hospital capacity, not only for the present, but projected into the future.

#### 2.4.10. Pre-Hospital Considerations

Only four articles were found relating pre-hospital considerations. Khorram-Manesh et al. [107] reviewed the regional registry at the Pre-hospital and Disaster Medicine Center in Sweden with respect to hospital-related incidents and its causes. Emergency department overcrowding, lack of beds at ordinary wards and/or intensive care units and technical problems at the radiology departments were the leading causes of the incidents, thus consuming capacity within the hospital. These incidents resulted in ambulance diversions and reduced pre-hospital capacity. Hanan et al. [108] undertook a robust evaluation of a community oncology nursing program in Ireland, which found that defined clinical procedures traditionally undertaken in hospitals were safely undertaken in the patient’s home with no adverse effects. The outcome of this program was a dramatic decrease in hospital attendances, thereby releasing hospital capacity. Patients also valued having care delivered at home and reported that it improved their quality of life, including reduced hospital visits and travel time. Burkett and Scott [109] focussed on the patients in aged care and developed a program that improved the quality of care for patients in a cost-effective way that minimised hospital admissions. These articles highlight the importance that pre-hospital considerations have on hospital capacity. Further research could include other novel ways to reduce hospital admissions, ‘no-show’ patients and length of stay at hospitals.

#### 2.4.11. Post-Hospital Considerations

One article was found considering post-hospital factors [110]. It addresses a vehicle scheduling problem encountered in home health care logistics. It concerns the delivery of drugs and medical devices, and pickup of bio samples, unused drugs, and medical devices from patients. Mixed-integer programming models are proposed and then Tabu Search and Genetic Algorithm heuristics are used because of computational intensity. While this problem was successfully solved for the given assumptions for a day, the authors acknowledged that in the real world, there are uncertainties around some variables. In addition, there is also a need for a much longer planning horizon than one day. These characteristics make the problem much more complex. To produce an implementable solution, perhaps other methods such as simulation optimisation may be required because it is well suited to encapsulating the manifest complexities within the system. The findings from this paper may be applied to other applications including the community oncology nursing program in Ireland described by Hanan, Mullen, Laffoy, O’Toole, Richmond and Wynne [108]. As technology increases, more services and tests may be performed outside the hospital thereby freeing up hospital capacity.

### 2.5. Analysis of Entire Hospital System Optimisation

Several articles were found that consider the entire hospital system and its capacity optimisation. Rechel et al. [111] discuss general principles and methodologies for hospital capacity planning and optimisation, but do not put forth any solution, method, or mathematical model for practical application. Hall [112] systematically goes through many aspects of hospital operations and discusses solutions but does not pull everything together into an integrated system. Yang et al. [113] built a multi-objective optimiser for three control factors—(1) bed allocation among wards, (2) overflow threshold (patients assigned to wards not ideally suited for their primary condition), and (3) discharge distribution—but these models did not have sufficient detail or include all the elements of a hospital to provide an accurate estimate of capacity. The authors of [2,114] developed a mixed-integer program to determine the maximum number of patients of each type that can be treated over time, or the time required to process a given cohort of patients. This work is significant because it quantified a patient throughput capacity for the major activities across the entire hospital, but its limitations were that it was a static model and did not account for stochasticity within and interaction between processes. Jones [115] puts forward a methodology for entire hospital optimisation, but it is static and does not quantify capacity, account for uncertainty, or consider the interdependencies of all components working together in real time. It acknowledged the limitations of LEAN and 6 Sigma. LEAN generally applies to manufacturing, where we can know and predict in detail what each of the parts will do in response to a given stimulus. In complex adaptive systems (e.g., hospitals), however, the “parts” have the freedom and ability to respond to stimuli in many ways [115]. While there may be application for LEAN and 6 Sigma methodologies to be employed in some sub-processes within a hospital, they are not suitable to be used as the only solution for entire hospital optimisation. The authors of [21] built a hospital-wide simulation model that facilitated the assessment of the impact of hospital-wide decisions and surge policies on each area of the hospital. The model did not include inventory, financials, maintenance, and ancillary staff. Bittencourt et al. [116] took a slightly different methodology and focussed on the role of queueing theory to hospital capacity management.

## 3. Discussion

### 3.1. Research Gaps

In general, there have been many studies on various parts of the hospital, with a major focus on core hospital activities, and a lack of studies on the ancillary parts which have a significant effect on entire hospital capacity optimisation. The other major limitation of the current research is that, in general, it lacks the detail required for practitioners and hospital administrators to implement solutions—Zhu, Fan, Yang, Pei and Pardalos [4]. In this section, we will focus on the research gaps of the two main areas that we believe require attention—architecture and operating theatre scheduling.

As alluded to earlier, architecture is immensely important to hospital capacity planning and optimisation because it directly impacts on every part of the hospital. For example, where should pharmaceutical outlets be located within the hospital, and how many should there be to minimise staff time procuring medications for patients? Where should the storerooms be located? What size and how many storerooms should there be to minimise staff time? Where should the shared resources such as imaging and laboratory departments be located to improve patient flow? Where should management staff be located so they can efficiently oversee the hospital operations? How many lifts and staircases should there be in the hospital to remove bottlenecks and improve patient and staff flow? Where should the green spaces and cafes be located so they enhance and not impede hospital and patient experiences? These are just a few research questions that represent a plethora of areas that need attention.

Operating theatre scheduling, on the other hand, has received enormous attention in the literature. Despite this, and perhaps due to the complex nature of the problem, this body of research has made surprisingly little impact in the real world (Zhu, Fan, Yang, Pei and Pardalos [4]). The task that we believe remains is to pull together all this research, into one model that can be practically implemented by hospitals. For example, the work by [79] was particularly noteworthy as it developed a unified resource model that can include specialised staff and medical equipment resource types (e.g., instrument nurse, surgical trays). The question remains as to how that approach can be scaled for a large hospital with over 1000 resource types and over 10,000 individual resources. Future research could also focus on the combination of interdependencies on various resources along with uncertainty in procedure times, emergency admissions and length of stay, within a rolling planning horizon, not just a static short-term horizon.

We suggest that a simulation-optimisation environment would be the best methodology to employ to address these major areas for further research. It is well known that these problems are computationally intractable (i.e., NP-hard), and as such, using meta-heuristics and/or deterministic approaches embedded within a simulation model that integrates all the parts of the hospital is ideal for addressing the architectural design questions and scheduling problems that remain unanswered.

### 3.2. Past and Future Research

Figure 4 and Figure 5 graphically portray the reviewed research over the last 20 years. A three-dimensional chart was chosen to illustrate the magnitude of articles considered in this review, with respect to the parts of the hospital.

When commenting on research trends, Madell, Villa, Hayward and Comte [89] observed that the increased computational complexity inherent with these types of questions explains the current trend of researchers to focus on deterministic approaches. Since the time that article was written, however, computer processing speeds have increased significantly. Ritchie [117] found that computing speeds by the largest supercomputer in any given year have increased from 93,000 trillion to 442,000 trillion (floating operations per second) from 2016 to 2020. This advance in technology, combined with the increase in data capture and availability, through evolved IT systems such as the integrated electronic medical record (iEMR) [118], has laid the foundation for more sophisticated and detailed modelling techniques to be employed. For example, the collection of timestamps of key events within a patient’s journey through the hospital [119] has lead to the inclusion of stochastic elements in models, giving rise to more accurate results and the ability to answer questions not previously attempted before. It is worth noting that exponential improvement in computing capacity only translates to linear improvement in the scale of NP-hard scheduling and optimisation problems that can be solved, so methods also need to be continually improved. Furthermore, Langabeer [120] states that the challenge for quality managers will be how to incorporate these new data into performance improvement programs and process changes for services that require attention. It is suggested that these are the predominant reasons accounting for the increasing trend for stochastic approaches in research papers compared to deterministic ones as seen in Figure 5. Many of the papers use both stochastic and deterministic approaches, generally using stochastic simulation to evaluate and test the deterministic approach [17,99,103,121,122,123,124,125,126,127,128].

## 4. Conclusions

This paper provides a timely and useful cross-section of operations research literature focussed on holistic optimisation of hospital capacity. It is particularly important that researchers continue to innovate, and that gaps between academic research and practical implementation are bridged. It is well known that, in general, demand for hospitals is exceeding capacity and widely accepted that there are still significant efficiency gains to be realised. The COVID-19 pandemic has brought this to light in an evocative way. This article provides an overview on hospital capacity optimisation and planning with 245 articles included for consideration. This review is novel in the sense that it summarises many parts of the hospital from an operations research perspective. A useful conceptual framework (see Figure 1) was also constructed to map the literature and research opportunities and to facilitate an understanding of the subject holistically, especially with respect to the strategic, tactical, and operational planning horizons. Perhaps the main finding, and most unexpected, was that despite the vast amount of published theoretical work on operating room management, there has been little or no impact of this work in an operational setting—Zhu, Fan, Yang, Pei and Pardalos [4].

A possible limitation of this study is the selection of articles to include in the review. Hospital capacity optimisation and planning are such broad and extensive subject, making it virtually impossible to include every article.

There are many useful techniques such as mixed-integer programming and meta-heuristics. Some are deterministic and some are stochastic, addressing one or more aspects of a hospital’s operation. However, it is unclear how these methods will perform in a dynamic, stochastic environment, where competition for resources occurs with other parts of the hospital, not included in the considered decision problem. Discrete event simulation is an established technique which can be used to test and calibrate these optimisation algorithms in such a challenging environment. It may also be used to consider how optimisation approaches focussing on different aspects of hospital operations can simultaneously work together. Furthermore, it is suggested that researchers focus on including more real-life elements in their problem descriptions so that the gap between research and the practical implementation of it is narrowed. For example, instrument trays and other essential equipment should be included in operating theatre schedule optimisation, but we could only find two articles that considered these [78,79].

As a final remark, it is hoped that this work will promote healthy, constructive discussions around hospital capacity in a holistic sense, and how it may be optimised when all the parts work together to achieve the ultimate end—quality of care. A guide to a holistic approach has been produced specifically for managerial staff to implement in their hospitals. The details may be found in Appendix B. Hopefully this article will also inspire others to develop solutions that can be implemented and have a profound impact in the real world.

## Figures and Tables

**Figure 1 healthcare-10-00826-f001:**
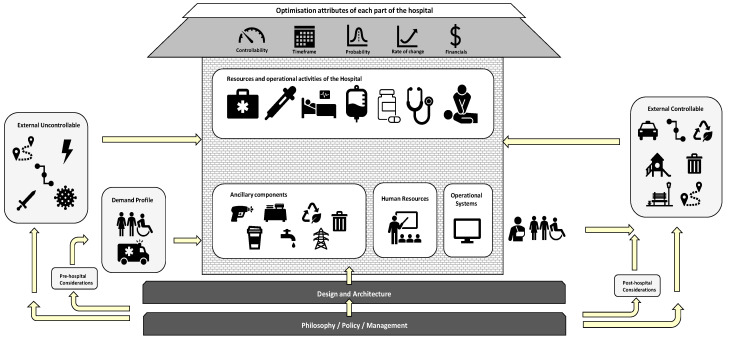
The ‘many parts, one body’ framework for understanding hospital capacity optimisation.

**Figure 2 healthcare-10-00826-f002:**
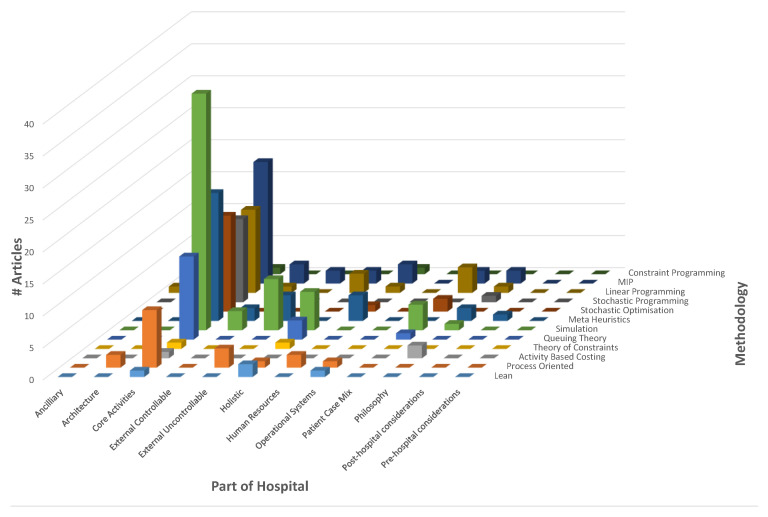
Literature review: prevalence of methodologies used by part of hospital.

**Figure 3 healthcare-10-00826-f003:**
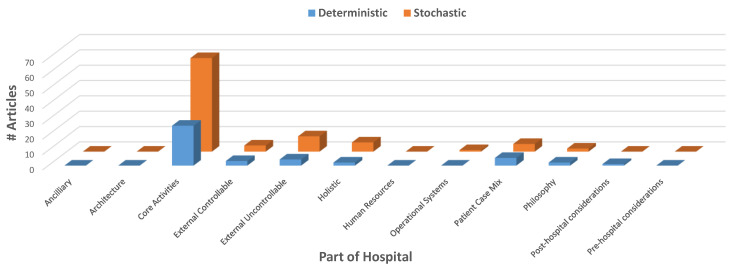
Literature review: problem characteristics by part of hospital.

**Figure 4 healthcare-10-00826-f004:**
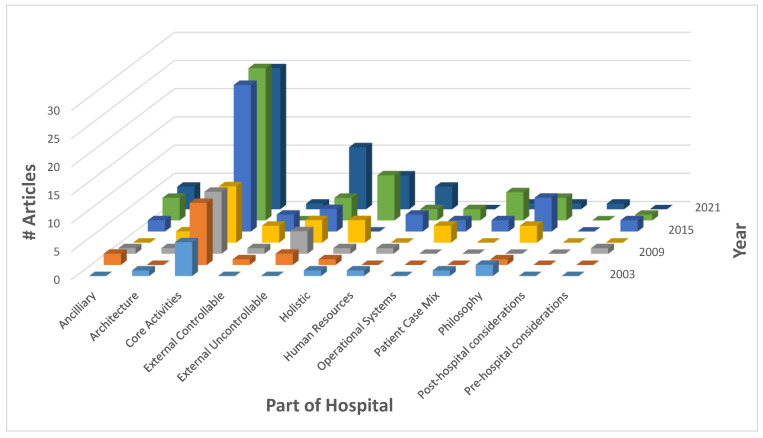
Number of articles reviewed by part of hospital by year.

**Figure 5 healthcare-10-00826-f005:**
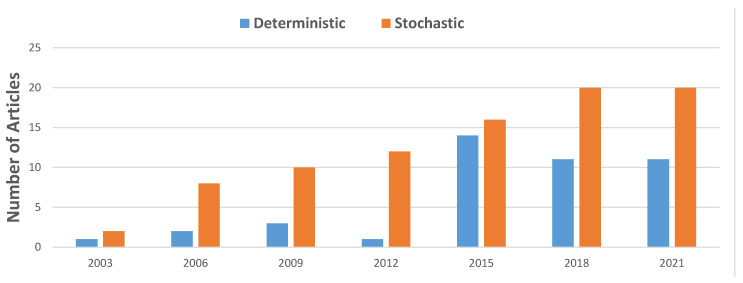
Number of articles by problem characteristics by year.

**Table 1 healthcare-10-00826-t001:** Literature search criteria and summary (Search history for the systematic review—Time period: 2000 to 2021).

				No. of Entries Returned in Search		
Part	Search Statement	Search Criteria	Date Searched	QUT Library	Google Scholar	Notes
Total subject	hospital capacity model	all words in the title.	Jun 2021	29	34	
Total subject	“hospital capacity”	all words in the title.	Jun 2021	230	984	
Total subject	Hospital capacity AND optim *	all words in the title.	Jun 2021	20	0	
Total subject	Hospital capacity AND simulation	all words in the title.	Jun 2021	25	42	
Total subject	Hospital capacity AND (mixed integer OR linear programming OR mixed integer linear programming)	all words in the abstract	Jun 2021	53	n/a	
Total subject	hospital AND (CAPEX OR ‘capacity planning’ OR ‘capacity analysis’ OR ‘capacity assessment’ OR ‘bottleneck analysis’ OR ‘work flow planning’ OR ‘system analysis’ OR ‘facility planning’ OR ‘patient pathways’ OR ‘capacity modelling’)	all words in the title	Jun 2021	74	n/a	Only 9 articles, all irrelevant, the rest newspaper articles.
Architecture	hospital capacity AND Architecture	all words in the title.	Jun 2021	0	0	
Architecture	hospital capacity AND Architecture	all words in the abstract	Jun 2021	1	n/a	
Architecture	hospital capacity AND Design	all words in the title.	Jun 2021	0	4	
Philosophy	hospital capacity AND (Philosophy OR Policy OR Management)	all words in the title.	Jun 2021	37	94	
Core Activities	hospital capacity AND (operations OR resources OR operating rooms OR operating theatres OR emergency department * OR bed * OR equipment OR scheduling OR resource consumption OR bed-planning OR theatre planning OR day surgery planning OR resource planning)	all words in the title.	Jun 2021	247	40	QUT Library: 194 of these were newspaper articles
Ancilliary	hospital capacity AND (food OR ‘patient food’ OR cafes OR maintenance OR waste OR trash OR recycling OR water OR wastewater OR sewerage OR electricity OR energy)	all words in the title.	Jun 2021	428	21	QUT Library: Only 3 of these were articles and 1 was a conference proceeding. The rest were newspaper articles.
Demand Profile	hospital AND (“casemix planning” OR “case-mix planning” OR “case mix planning” OR “demand profile” or patient case mix)	all words in the abstract	Jun 2021	7	n/a	
External Uncontrollable	hospital capacity AND (‘natural disasters’ OR earthquakes OR floods OR fires OR pandemics OR war OR location OR ‘supply chains’)	all words in the title.	Jun 2021	1	15	
External Uncontrollable	hospital capacity AND (‘natural disasters’ OR earthquakes OR floods OR fires OR pandemics OR war OR location OR ‘supply chains’)	all words in the abstract	Jun 2021	31	n/a	mostly not relevant
External Controllable	hospital capacity AND (car parks OR green spaces OR playgrounds OR waste OR electricity or wastewater or sewerage or location or recycling or supply chains)	all words in the title.	Jun 2021	0	3	Only related to location
Human Resources	“hospital capacity” AND (training OR “professional development” OR communication OR safety OR error * OR teamwork OR staff)	all words in the abstract	Jun 2021	109	5	
Human Resources	nurse rostering problem	all words in any field	Jun 2021	1998	17,000	
Pre-hospital considerations	hospital capacity AND (prevention)	all words in the title.	Jun 2021	0	0	
Post-hospital considerations	“hospital capacity” AND (“outpatient care” OR “prevention of re-admission”)	all words in the title.	Jun 2021	0	0	
Operational Systems	hospital capacity AND (IT systems OR payroll systems OR scheduling software OR scheduling tools OR planning tools)	all words in the title.	Jun 2021	0	4	Google scholar: irrelevant articles
**Totals**				**3290**	**18,246**	

**Notes**: the asterisk indicates a wildcard. It will return all the words that begin with the letters prior to the asterisk. Eg. Optim * includes optimisation, optimal, optimising etc.

**Table 2 healthcare-10-00826-t002:** Summary of articles by part, sub-part and problem-solving approach.

	#	Stoch.	Det.		#	Stoch.	Det.
**Ancilliary**	**13**			**Holistic**	**21**	**6**	**2**
Cooling, Heating and Power	3			Assessment Tool	1		
Energy	1			Hospital Optimisation	9	1	2
Energy, Heating	1			Literature Review	2		
Facility Management	3			Policy	1		
IT Systems	1			Surge events	1	1	
Safety	1			Capacity Management	1		
Staff	1			Core Activities	1	1	
Training	1			No secondary category	5	3	
Wastewater	1			**Human Resources**	**11**		
**Architecture**	**5**			Literature Review	1		
Facility Management	3			Nursing Staff	8		
Safety	1			Safety	1		
No secondary category	1			No secondary category	1		
**Core Activities**	**118**	**61**	**26**	**Operational Systems**	**7**	**1**	
Beds	29	18	11	IT Systems	6	1	
Capacity Planning	1			Medical Records	1		
Capacity Strain	1			**Patient Case Mix**	**9**	**5**	**5**
Communication	1			Literature Review	1		
Emergency	15	10	1	Operating Rooms	1	1	
Inventory	2			No secondary category	7	4	5
Literature Review	7	2		**Philosophy**	**17**	**2**	**2**
Nursing Staff	1			Core Activities	1		
Operating Rooms	5	1	1	Decision Making	2		
Patient flow	3	3		Financials	4	2	1
Post-hospital considerations	1			IT Systems	1		
Pre-hospital considerations	1			Literature Review	3		
Process Flow	1	1		Patient Case Mix	1		
Resources	9	5		Policy	2		
Scheduling	28	14	12	No secondary category	3		1
Staff	2	1	1	**Post-hospital considerations**	**1**		**1**
Surge events	1	1		No secondary category	1		1
No secondary category	10	5		**Pre-hospital considerations**	**4**		
**External Controllable**	**9**	**4**	**3**	Post-hospital considerations	1		
Location	6	3	1	Prevention	1		
Multi-Hospital Network	2		2	No secondary category	2		
Supply Chain	1	1		**Grand Total**	**245**	**89**	**43**
**External Uncontrollable**	**30**	**10**	**4**				
Core Activities	1	1					
Literature Review	2						
Pre-hospital considerations	1						
Surge events	25	9	4				
No secondary category	1

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
