# Peer review of "An Overview of Hospital Capacity Planning and Optimisation"

_healthcare, 2022, doi:10.3390/healthcare10050826_

Round 1

Reviewer 1 Report

Thank you for the opportunity to review this manuscript. This research attempted to provide an overview of the optimisation of hospital capacity and planning, and its focus will be to take a detailed view of hospital capacity, mapping out its various components. While I believe the topic (as suggested by the purpose) is an important one, I am not sure if the author has adequately/accurately addressed the study aims. 

First of all the abstract is supposed to be the summary of the main research idea. Here the abstract is more like introduction, that should be corrected. Research question should be highlighed, please refer to some journal papers in the area and see how an abstract is structured.

In addition to analysing the literature, interviews with staff in key roles within South-East Queensland hospitals were conducted. There is no detailed description of the step and result of interviews.

Authors mentioned “a search of the top medical journals was also conducted to ensure the literature review was targeted to the industry”. Please list the searching strategies, searching keywords, selecting database. As to Boolean operators, please explain how these keywords were screened out.

Reviewer 2 Report

The paper is all fine in all key aspects of scientific reports, and publishable as it is.  However, I think the good paper would extremely benefit from addressing some key issues.

As ever, the manuscript leaves some items open for discussion and improvement.  I ask the authors to consider the following issues when finalizing the article:

A key issue I wonder very much is your sticking to the terms stochastic and deterministic. Is there no room for qualitative research in your science understanding, is mathematical modelling the only acceptable way to do research. In addition to keeping these concepts as self-clear, you do not discuss this issue at all, just keep it as given.  This part needs further exploration. You yourself say this very late int he article: "Hospital capacity optimisation is much more than solving a mathematical programming problem because it involves the subjective factors in addition to the objective ones".  Why is this not reflected and adhered to when the articles are analyzed?

There is no decent description why the articles were selected from gut library and Google Scholar.  Especially gut library is a very vague term, its surely offered gateways to many established scientific databases.  This should be explained in more detail. In addition, defining gut library as the source means that the research is not possible to repeat and track for anyone outside the user group of qut library.

These are understatements: "Many hospitals in the world are constrained by resources and many people in the world have limited funds to avail themselves of those services"  You could say exactly that all hospitals and all people.  According to basic theories of economics, for each economic entity some resource is scarce.

In the first paragprah of 1.2 you have repetition.

Again an understatement:  "As this literature review suggests, the healthcare sector is not yet optimised".  Few things in the world will ever be optimised, healthcare as a total entity never.

Somehow your article leaves an unbalanced feeling.  You for example discuss very scarcely medicine, blood and spare organ supply to hospitals, all which are key operating problems,  Shortage of IS is also weakly discussed. In the other side, you go deep to trivial (in totality) problems such as "surgical tray configuration".

A key issue that you neglect wholly is the availability of patients.  No-show patients are maybe a key hazard to hospital productivity.  I wonder why did you not come up with this issue not at all in the literature review.  I think this issues should be discussed in every case to some extend.

You also refer to parking space needed.  I think in this connection the reachability of hospitals by public transportation should as well be discussed.

You use in some places odd bindings of words, so the reader is somewhat lost:

"Inventory issues, management and policies within hospitals are generally not considered in hospital optimisation studies" Is this three issues, inventory,  management and policies, or about management and policies about inventory.    

In the same way: "No articles were found for hospital capacity optimisation and cleaning". Also no articles for cleaning and neither for capacity optimization, or no articles about capacity optimization of cleaning?

This discussion seems like superficial on not needed:  "Hinic, et al. [63] describes an academic-practice partnership that has increased the hospital's capacity for research, evidence-based practice, and support for nurses continuing their education. "  Is this not the principle that university hospitals have lived with for long times already?

This section is rather blurring:

"Ritchie [118] found that computing speeds by the largest supercomputer in any given year have increased from 93,000 trillion to 442,000 trillion (floating operations per second) from 2016 to 2020. This advance in technology, combined with the increase in data capture and availability, through evolved IT systems such as the integrated electronic medical record (iEMR)[119], has laid the foundation for more sophisticated and detailed modelling techniques to be employed to obtain more accuracy and find answers to questions not previously been able to solve before. "   These issues are very little connected, and especially electronic medical records structures have very little to do with sophisticated and detailed modelling.

Discuss what you mean when saying "problems are NP-hard"

When discussing nurse rostering problem, you could maybe refer to the other term widely used: nurse scheduling problem.

I think you give all too little value to medication when you call it ancillary task: "One article that did highlight the importance of ancillary tasks such 
as medication delivery and lab sample collection".  I would say that majority of hospital cases are handled through medicines, with no interventions, so medication is at the very core of care, cure and hospital activities.   I think you should address this in the whole article, including figure 1

In page 6 you refer to appendix 0.

This definition is rather misleading:   "a large hospital with 5,000 resources". A hospital might well already have 5 000 employees, or there are easily over 10 000 different medicines available in the hospital pharmacy. Are you talking about resource types, not about individual resources?

In page 7 spelling mistake:  " It also highlights that stochastic approaches are employed more abundantly than deterministic ones For the purposes of this classification, stochastic refers to approaches that consider random"

In first page I would write capacity-related, not capacity related (which would be in line with the rest of the use of the word in the article)

I find this sentence unclear and badly structured:  "Secondly, the consideration of competition as financial pressure rises, and hospital systems are opened to the market" (page 15).

Again a statement that you undervalue important things:  "The rest discuss static analysis of costs, that are not intricately related to the operations of the hospital ". As in any human activity, cost is always a major issue intricately affecting human and organizational operations.  Saying that costs do not affect hospital operations is just lying.

In page 16, it is not necessary to have the book name mentioned. As always, details belong to the reference list.

At the end we have a serious overstatement:  "This review is novel in the sense that it summarises all the parts of the hospital from an operations research perspective."  You do not summarize all parts of the hospital, I doubt anyone will ever do.

You could to some more extent discuss the term "overflow threshold".  It is so widely used in so many areas that some definition would needed.  Overflow of what and where?

Be needed?: "Implementation of these solutions have the potential to save a significant amount of money which can BE used in other areas of the hospital, while at the same time reducing supply risks of these critical resources."

Round 2

Reviewer 1 Report

It seems that the authors have tried their best to revise the manuscript.